# ASML: A Scalable and Efficient AutoML Solution for Data Streams

**Nilesh Verma**[1,*]  **Albert Bifet**[1,*]  **Bernhard Pfahringer**[1,*]  **Maroua Bahri**[2,*]

[1]AI Institute, University of Waikato
[2]Inria PARIS, France
[*]Equal contribution.

**Abstract**  Online learning poses a significant challenge to AutoML, as the best model and configuration may change depending on the data distribution. To address this challenge, we propose Automated Streaming Machine Learning (ASML), an online learning framework that automatically finds the best machine learning models and their configurations for changing data streams. It adapts to the online learning scenario by continuously exploring a large and diverse pipeline configuration space. It uses an adaptive optimisation technique that utilizes the current best design, adaptive random directed nearby search, and an ensemble of best performing pipelines. We experimented with real and synthetic drifting data streams and showed that ASML can build accurate and adaptive pipelines by constantly exploring and responding to changes. In several datasets, it outperforms existing online AutoML and state-of-the-art online learning algorithms.

## 1 Introduction

Data streams are unbounded data sequences that arrive continuously and may change over time. They are generated by various devices such as IoT sensors, mobile phones, and websites and have applications in many domains such as transportation, finance, healthcare, and more (Shinde and Shah, 2018). Analyzing these real-time data streams is crucial for extracting insights and making decisions (Lu et al., 2018). Applying Machine learning (ML) to real-world problems requires multiple complex steps such as data preprocessing, feature engineering, model selection, and hyperparameter tuning, which demand significant expertise and time from humans (Gijsbers and Vanschoren, 2021). To make ML more accessible to non-experts, automated machine learning (AutoML) has emerged to automate the end-to-end ML pipeline (Hutter et al., 2015).

While AutoML has achieved great success with static or offline datasets, applying it to dynamic or online data streams poses new challenges. ML models must continuously adapt to the changing data distribution as new data arrives (Bifet et al., 2018). To address these challenges, we propose Automated Streaming Machine Learning (ASML) - an automated machine learning framework tailored for data streams. The main contributions of this work are:

- A novel AutoML formulation for online learning on evolving data streams.

- An adaptive random-directed nearby search strategy that continuously explores the pipeline configuration space to find optimal solutions over time.

- An experimental study that shows its effectiveness and resource efficiency over existing Online AutoML and state-of-the-art online learning algorithms.

The rest of the paper is organized as follows: section 2 provides related work on AutoML in data streams; section 3 presents the problem formulation; section 4 describes the proposed ASML framework; section 5 reports the experimental setup; section 6 provides results and discussions; and section 7 concludes the paper and presents future works.

## 2 Related Work

Automated machine learning (AutoML) tackles the Combined Algorithm Selection and Hyperparameter (CASH) optimization problem (Feurer et al., 2015), which involves finding the best combination of ML algorithms and their hyperparameters for a given task. Several AutoML frameworks have been developed to handle various aspects of the ML pipeline, such as data preparation, feature engineering, model selection, hyperparameter optimization, and ensembling. Some examples of these frameworks are Auto-Weka (Thornton et al., 2013), Auto-Sklearn (Feurer et al., 2022), TPOT (Olson and Moore, 2016), H2O.ai (LeDell and Poirier, 2020), GAMA (Gijsbers and Vanschoren, 2021), AutoGluon (Erickson et al., 2020), and FLAML (Wang et al., 2021). AutoML frameworks employ various search methods to efficiently and effectively find the optimal configurations, such as Grid Search (LaValle et al., 2004), Random Search (Bergstra and Bengio, 2012), Bayesian Optimization (Snoek et al., 2012), Tree-structured Parzen Estimator (TPE) (Bergstra et al., 2011), Population-Based Methods (Cheng et al., 2016) (e.g., Genetic Algorithms (De Jong, 1988)), and Bandit-based Approaches (Li et al., 2018). While AutoML frameworks have greatly accelerated model development for static datasets, they still face challenges when applied to data streams.

Machine learning (ML) algorithms operating on data streams must adapt to dynamic environments while facing limited computing, memory, and time resources per sample (Bifet et al., 2011). A key challenge in data stream ML is concept drift, which occurs when the underlying data distribution changes over time (Bifet et al., 2011). Concept drift can be abrupt, gradual, incremental, or recurring (Lu et al., 2018). Several techniques have been proposed to handle concept drift in data streams, such as HAT (Hoeffding Adaptive Tree) (Hulten et al., 2001), Leveraging Bagging (Bifet et al., 2010), ARF (Adaptive Random Forest) (Gomes et al., 2017), and SRP (Streaming Random Patches) (Gomes et al., 2019). These are popular machine learning algorithms for data streams that use various mechanisms to detect and adapt to concept drift, such as error-rate tracking, change detection, and ensemble updates (Lu et al., 2018). The most common evaluation method for data stream ML is prequential evaluation, which tests and trains models on each new instance (Bifet et al., 2015). However, these techniques still require expert knowledge to be configured properly for the data stream environment.

Recent research has started exploring AutoML tailored to data streams. Initial works like Online AutoML (OAML) (Celik et al., 2023) use asynchronous evolutionary search in combination with explicit drift detection to optimize ML pipelines in data stream settings. Evolution-based Online Automated Machine Learning (EvoAutoML) (Kulbach et al., 2022) maintains a population of heterogeneous pipeline configurations, evolving them using evolutionary random mutations. AutoClass (Bahri and Georgantas, 2023) is an automated machine learning method for data stream classification that uses an ensemble of configurations to dynamically select a parent configuration proportional to its accuracy using roulette wheel selection and generates a new child configuration by mutating the parent's hyperparameters via truncated normal distributions for numeric values and sampling probabilities for categorical values. The MetaStream (Rossi et al., 2014) method employs meta-learning to select single ML algorithms or ensembles based on extracted statistical features from a data stream. Methods like SSPT (Veloso et al., 2021) focus on efficient hyperparameter tuning in the stream using the Nelder-Mead optimization algorithm, and ChaCha (Champion-Challengers) (Wu et al., 2021) employed hyperparameter tuning in resource-constrained settings for regression tasks. A survey paper (Imbrea, 2021) compares classic batch models, online learning algorithms, pre-trained AutoML pipelines, and online meta-learning for algorithm selection in the context of data streams, underlining the importance of adaptation to concept drift for effective AutoML.

However, some of these works have limitations in terms of search space, adaptation capabilities, feature engineering support, and evaluation rigour. To address these gaps, more comprehensive systems are needed that leverage inherent online learning algorithms within optimized and con-

figurable pipelines and are evaluated on real-world non-stationary streams. Our proposed ASML system aims to address these gaps.

## 3 Problem Formulation

In this section, we formally define the problem of online automated machine learning for data streams. Given an unbounded data stream $\mathcal{D} = (x_t, y_t)$, where $x_t$ represents the input features at time step $t$ and $y_t$ denotes the target labels at time step $t$, the goal is to find the optimal combination $S$ of data preprocessing $p$, input feature $f$, machine learning classifier $c$, and hyperparameter $h$ that maximizes a defined objective function $\mathcal{O}$ over the evolving data stream $\mathcal{D}$. The search process is guided by an algorithm $\mathcal{A}$ that adapts model optimization based on detected concept drift in the stream. Formally, the Online AutoML optimization problem is:

$$S_{\text{best}}^t = \underset{(p,f,c,h) \in P \times F \times C \times H}{\operatorname{argmax}} \mathcal{O}_t(\mathcal{A}\{\mathcal{D}_t\}) \tag{1}$$

Where $P$, $F$, $C$, and $H$ represent the spaces of possible configurations for the preprocessing, features, models, and hyperparameters, respectively.

Additionally, in the scenario of data streams, we address the Online CASH (Combined Algorithm Selection and Hyperparameter optimization) problem, focusing on identifying the optimal configuration for machine learning pipelines at each time step $t$. This optimal configuration is represented as $A_{\lambda,t}^*$, which is chosen from a set of online learning algorithms $A_{OL}$:

$$A_{\lambda,t}^* = {}^*\operatorname{argmin}_{\forall A^j \in A_{OL}} L\left(A_{\lambda,t}^j, \{x_t, y_t\}\right) \tag{2}$$

The goal is to minimize the loss function $L$, which evaluates the performance of an algorithm $A^j$ with a specific set of hyperparameters $\lambda$ using the most recent data stream $\{x_t, y_t\}$.

## 4 Automated Streaming Machine Learning (ASML)

In this paper, we propose ASML, a new method that automatically adapts to a data stream environment. It solves the online CASH problem (Feurer et al., 2015) that we defined in section 3. At the moment, it can only handle classification tasks, but we can easily extend it to other types of supervised learning problems in the future. It uses the base classifiers from the online learning library River (Montiel et al., 2021).

ASML tries to find the best pipeline configuration by exploring and changing it continuously. A pipeline has three parts: pre-processors to change the input data, feature selectors to choose the most useful features, and classifiers to make predictions. Each part has some hyperparameters that affect how it works. We show the ASML high-level diagram in Figure 1 and its working steps in Algorithm 1.

ASML searches for the best pipeline configuration for streaming data. A pipeline consists of components and their hyperparameters. ASML evaluates all possible pipelines on the first $W$ examples using prequential evaluation Bifet et al. (2015) and selects the best one $s_{\text{best}}$.

Then, for every new batch of $W$ examples, ASML explores $B$ pipelines in parallel, where $B$ is the budget. The $B$ pipelines comprise the current best pipeline $s_{\text{best}}$, $(B-1)/2$ pipelines generated by Adaptive Random Directed Nearby Search (ARDNS), and $(B-1)/2$ pipelines generated randomly from the search space. ARDNS randomly changes the hyperparameters of the current best pipeline by choosing one of four directions: Same ($s$), Upper ($u$), Lower ($l$), or Random ($r$); see Figure 2 and Algorithm 2 for details. This function explores the hyperparameter space around the current best pipeline and increases the chance of finding better solutions. The random function adds diversity to the search and helps to adapt to changes in the data distribution.

ASML has two modes of prediction: Ensemble prediction (ASML_E) and Best model prediction (ASML_B). In Ensemble mode, ASML maintains an ensemble model $E$ of the best pipelines selected

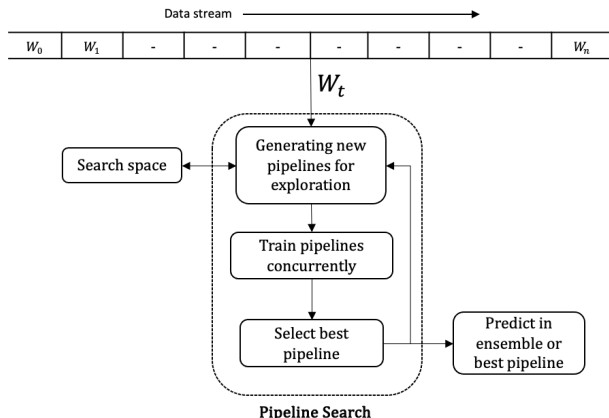

Figure 1: ASML high-level process of new pipeline search, where $W_t$ represent the data stream window in time step $t$.

---

**Algorithm 1** ASML (Ensemble)

---

**Input:** Preprocessors $P$, Feature selectors $F$, Classifiers $C$, Hyperparameter space $H$, Exploration window size $W$, Budget $B$, Metrics $M$, Ensemble model $E = \{e_1, e_2, \ldots, e_n\}$ of size $n$ and data stream instances $\mathcal{D} = (x_t, y_t)$

**Output:** Best configuration $S_{\text{best}}$

1: $S_{\text{init}} \leftarrow P \times F \times C$
2: $B_{\text{ARDNS}} \leftarrow (B-1)/2$        ▷ Count of new pipelines generated by ARDNS
3: $B_{\text{Random}} \leftarrow B - B_{\text{ARDNS}}$        ▷ Count of new pipelines generated randomly
4: $t \leftarrow 0$
5: **while** $(x_t, y_t) \in$ first window of size $W$ **do**        ▷ Initialization
6:     **for** each $s \in S_{\text{init}}$ **do**
7:         Predict $\hat{y}_t \leftarrow s(x_t)$        ▷ Prequential evaluation
8:         Evaluate $M_t \leftarrow M_t(\hat{y}_t, y_t)$
9:         Train $s(x_t, y_t)$
10:     **end for**
11: **end while**
12: $S \leftarrow S_{\text{init}}$
13: **while** $(x_t, y_t) \in \mathcal{D}$ **do**
14:     **if** $t \mod W == 0$ **then**
15:         $s_{\text{best}} \leftarrow s_{argmax(M)}$        ▷ Select pipeline with the best metric score
16:         **if** $|E| == n$ **then**
17:             $e_{\text{worst}} \leftarrow e_{argmin(e_M)}$
18:             Remove $e_{\text{worst}}$ from $E$        ▷ Remove worst pipeline from ensemble pool
19:         **end if**
20:         Add $s_{\text{best}}$ to $E$        ▷ Add best pipeline to ensemble pool
21:         $S_{\text{ARDNS}} \leftarrow \text{ARDNS}(s_{\text{best}}, H, B_{\text{ARDNS}})$        ▷ New pipelines via ARDNS(Algorithm 2)
22:         $S_{\text{Random}} \leftarrow \text{Random}(S_{\text{init}}, B_{\text{Random}}))$        ▷ New random pipelines
23:         $S \leftarrow s_{\text{best}} \cup S_{\text{ARDNS}} \cup S_{\text{Random}}$        ▷ Update pipeline set
24:     **end if**
25:     Train $S(x_t, y_t)$        ▷ Train pipelines concurrently
26:     Train $E(x_t, y_t)$        ▷ Train ensemble
27:     $t \leftarrow t + 1$
28: **end while**

---

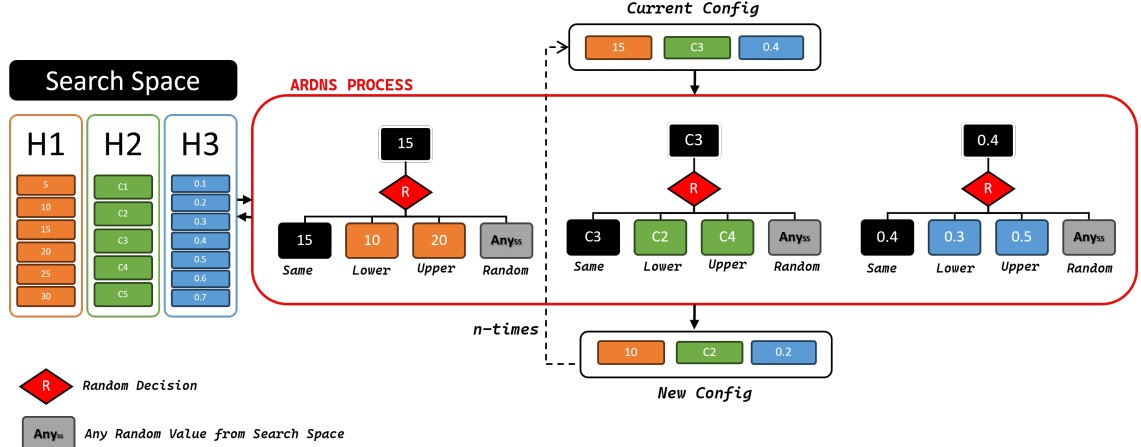

Figure 2: The ARDNS process. H1, H2, and H3 represent hyperparameter values in the search space. Parameters from H1 and H2 are chosen from a lower direction, while H3 is picked randomly. The process is repeated **n** times, generating **n** new configurations.

---

**Algorithm 2** ARDNS Method

---

**Input**: Hyperparameter values of the best pipeline $\theta$, Hyperparameter space $H$.
**Output**: New Hyperparameter values $\theta^*$

 1:  $d \leftarrow \text{Random}(s, u, l, r)$                    ▷ Randomly select direction
 2:  **if** $d = s$ **then**                           ▷ Same direction
 3:    $\theta^* \leftarrow \theta$
 4:  **else if** $d = u$ **then**                       ▷ Upper direction
 5:    $\theta^* \leftarrow H[\min(\text{index}(\theta, H) + 1, \ \text{length}(H) - 1)]$
 6:  **else if** $d = l$ **then**                       ▷ Lower direction
 7:    $\theta^* \leftarrow H[\max(\text{index}(\theta, H) - 1, \ 0)]$
 8:  **else if** $d = r$ **then**                      ▷ Random direction
 9:    $\theta^* \leftarrow \text{Random}(H)$
10:  **end if**
11:  **return** $\theta^*$

---

after each batch of $W$ examples. The ensemble size $n$ is fixed at the beginning. Each pipeline in the ensemble has its own online metric $e_m$ to measure its prequential performance. If the ensemble is full ($|E| == n$), the pipeline with the lowest $e_m$ is removed. ASML_E predicts using a confidence-weighted majority vote of the ensemble. In the Best mode, ASML uses only the best pipeline $s_{\text{best}}$ from the last batch of $W$ examples to make predictions for the new data instances.

## 5 Experimental Setup

This section describes the search space, datasets, baselines, and experimental configuration for ASML.

  **Search Space**: The search space is a fundamental element of any AutoML system. For our search space, we have chosen six classifiers: Perceptron, Logistic Regression (LR), Gaussian Naive Bayes (GaussianNB), Hoeffding Tree Classifier (HAT), K-Nearest Neighbors (KNN) Classifier, and Adaptive Random Forest Classifier (ARF). Additionally, we have included two preprocessing algorithms, MinMaxScaler and StandardScaler, to normalize the data. Three algorithms are used for feature engineering to select the best features based on statistics: Poisson Inclusion, SelectKBest, and Variance Threshold. Each algorithm has specific hyperparameters that can be adjusted for optimal

performance in different online learning scenarios. The algorithms and their hyperparameter details are shown in Table 1.

Table 1: Hyperparameters search space for various data stream algorithms from River (Montiel et al., 2021).

| Algorithm | Hyperparameter | Search Range | Steps | Type |
|---|---|---|---|---|
| Perceptron | l2 | 0.00 to 0.01 | 0.001 | float |
| LogisticRegression | l2 | 0.00 to 0.01 | 0.001 | float |
| GaussianNB | | | | |
| HoeffdingTreeClassifier | max_depth | 10 to 100 | 10 | int |
| | grace_period | 50 to 500 | 50 | int |
| | split_confidence | 1e-9, 1e-7, 1e-4, 1e-2 | 0.1 | float |
| | tie_threshold | 0.02 to 0.08 | 0.01 | float |
| | nb_threshold | 0 to 50 | 10 | int |
| | split_criterion | info_gain, gini, hellinger | | cat |
| | leaf_prediction | mc, nb, nba | | cat |
| AdaptiveRandomForestClassifier | n_models | 2 to 9 | 1 | int |
| | max_depth | 10 to 100 | 10 | int |
| | grace_period | 50 to 500 | 50 | int |
| | lambda_value | 2 to 9 | 1 | int |
| | split_confidence | 0.01 to 0.1 | 0.01 | float |
| | tie_threshold | 0.02 to 0.08 | 0.01 | float |
| | nb_threshold | 0 to 50 | 10 | int |
| | split_criterion | info_gain, gini, hellinger | | cat |
| | leaf_prediction | mc, nb, nba | | cat |
| KNNClassifier | n_neighbors | 3 to 8 | 1 | int |
| | window_size | 100 to 5100 | 200 | int |
| | weighted | True, False | | bool |
| | p | 1 to 4 | 1 | int |
| MinMaxScaler | | | | |
| StandardScaler | with_std | True, False | | bool |
| PoissonInclusion | p | 0.1 to 1.0 | 0.1 | float |
| VarianceThreshold | threshold | 0.0 to 1.0 | 0.1 | float |
| | min_samples | 1 to 9 | 1 | int |
| SelectKBest | k | 1 to 25 | 1 | int |
| | similarity | PearsonCorr, Cov | | cat |

**Datasets**: To evaluate the performance of different methods in terms of prequential accuracy, time (Sec.), and memory (M.B.) usage, we used 14 datasets that cover various streaming challenges, as shown in Table 2. These datasets include both artificial (synthetic) and real-world data.

**Baselines and Experimental Configuration**: We experimented with two versions of the proposed ASML method: an ensemble version (ASML_E) and a best model selection version (ASML_B). We compared them with AutoClass (Bahri and Georgantas, 2023), EvoAutoML (EAML) (Kulbach et al., 2022), Online AutoML (OAML) (Celik et al., 2023), and state-of-the-art online algorithms such as Hoeffding Adaptive Tree Classifier (HATC) (Bifet et al., 2018; Hulten et al., 2001), Adaptive Random Forest classifier (ARFC) (Gomes et al., 2017) and Streaming Random Patches ensemble classifier (SRPC) (Gomes et al., 2019). All algorithms were evaluated using their default configurations, as specified in their original papers. We applied the configuration of ASML, which consists of an online accuracy metric, an ensemble of size 3 in an ensemble prediction mode, an exploration

Table 2: Datasets used in the evaluation.

| Index | Dataset | Features | Classes | Instances | Type |
|------:|---------|---------:|--------:|----------:|------|
| 1 | Adult (Becker and Kohavi, 1996) | 14 | 4 | 48,842 | Real World |
| 2 | Electricity (Harries et al., 1999) | 8 | 2 | 45,312 | Real World |
| 3 | Forest Cover (Blackard, 1998) | 54 | 7 | 581,012 | Real World |
| 4 | Insects (Souza et al., 2020) | 33 | 6 | 52,848 | Real World |
| 5 | New Airline (Bifet et al., 2011) | 7 | 2 | 500,000 | Real World |
| 6 | Shuttle (Newman et al., 1998) | 9 | 7 | 58,000 | Real World |
| 7 | Vehicle SensIT (Duarte and Hu, 2004) | 100 | 3 | 98,528 | Real World |
| 8 | HYPERPLANE - High Gradual Drift (Hulten et al., 2001) | 10 | 2 | 500,000 | Synthetic |
| 9 | Moving RBF (Bifet et al., 2011; Losing et al., 2016) | 10 | 5 | 200,000 | Synthetic |
| 10 | Moving Squares (Losing et al., 2016) | 2 | 4 | 200,000 | Synthetic |
| 11 | SEA - High Abrupt Drift (Hulten et al., 2001) | 3 | 2 | 500,000 | Synthetic |
| 12 | SEA - High Mixed Drift (Hulten et al., 2001) | 3 | 2 | 500,000 | Synthetic |
| 13 | Random RBF Drift (Montiel et al., 2021) | 4 | 4 | 100,000 | Synthetic |
| 14 | Agrawal (Montiel et al., 2021) | 9 | 2 | 100,000 | Synthetic |

window of size 1000, and a budget of 10. We repeated the experiments with different random seeds, ranging from 42 to 52. The computational tests were performed on a Linux server with an AMD EPYC 7702 64-core Processor and approximately 1 TB of RAM.

# 6 Results

In this section, we present the results of our experiments. Our results showed that both versions of ASML outperformed the other methods in all aspects. ASML_E and ASML_B achieved average accuracy scores of 86.05% and 85.00%, respectively. This was much higher than the 81.39% accuracy of AutoClass, 83.84% accuracy of EAML, and 80.91% accuracy of OAML as shown in Table 3. ASML_B was also the fastest method, with an average runtime of 905.81 seconds, compared to 3317.29 seconds for AutoClass, 2502.05 seconds for EAML, and 3432.91 seconds for OAML as shown in Table 4 and Figure 3a. ASML_E took more time at 1720.47 seconds on average but achieved the highest accuracy. Moreover, both versions of ASML used much less memory than the other methods. ASML_E consumed 25.63 MB, and ASML_B consumed 22.80 MB on average, while EAML used 190.89 MB, OAML used 63.34 MB, and AutoClass used 34.95 MB on average, as shown in Table 5 and Figure 3b. For reference, the online algorithms ARFC, SRPC, and HATC achieved average accuracies of 77 to 78%, run times of 300 to 1300 seconds, and memory usage between 15 to 58 MB. However, they were significantly inferior to the Online AutoML techniques.

In summary, ASML_E traded some extra computational resources for a 1 to 2% increase in accuracy over ASML_B. Depending on the use case and available resources, either the ensemble approach can be used, or alternatively ASML_B also offers a good balance between predictive performance and efficiency. Notably, both versions of ASML achieved state-of-the-art results compared to the existing AutoML techniques. ASML_E set a new record for the highest accuracy for AutoML on the datasets we evaluated, while ASML_B also surpassed the previous methods considerably while requiring fewer computational resources.

## 6.1 Statistical Comparison

We used the Nemenyi non-parametric statistical test to compare the average ranks of multiple algorithms. We compared ASML_E and ASML_B with other AutoML and Online machine learning algorithms based on accuracy, execution time, and memory usage, with a 95% confidence level. The accuracy comparison (Figure 4a) shows that ASML_E and ASML_B have higher ranks compared to the baselines methods but are statically similar to the EAML, OAML, ARFC, and AutoClass, meaning that they are not significantly different in terms of accuracy. However, the execution time

Table 3: Accuracy comparison.

| Datasets | ASML_E | ASML_B | AutoClass | OAML | EAML | ARFC | SRPC | HATC |
|---|---|---|---|---|---|---|---|---|
| Adult | 80.01±0.61 | 80.36±0.27 | 76.86±1.28 | 72.07±0.44 | 80.56±1.29 | 81.35±0.33 | 80.11±0.30 | **81.64±0.27** |
| Electricity | **91.50±0.12** | 90.65±0.26 | 87.98±1.16 | 86.96±0.49 | 89.13±0.44 | 85.84±0.25 | 86.63±0.16 | 83.19±0.32 |
| Forest Cover | **95.63±0.07** | 95.39±0.18 | 95.32±0.02 | 83.16±0.52 | 94.07±0.06 | 88.85±0.40 | 92.97±0.09 | 70.99±1.48 |
| Insects | 70.95±0.46 | **71.25±0.28** | 64.24±0.25 | 63.69±0.27 | 70.05±1.66 | 68.61±0.48 | 68.60±0.51 | 60.25±1.50 |
| New Airlines | 66.58±0.09 | 65.46±0.05 | 63.03±0.48 | 67.03±0.49 | **67.64±0.34** | 65.31±0.09 | 64.53±0.17 | 65.27±0.10 |
| Shuttle | 99.34±0.07 | 98.58±0.11 | **99.66±0.03** | 97.31±0.18 | 98.65±0.25 | 99.54±0.07 | 99.48±0.08 | 94.57±0.69 |
| Vehicle Sensit | **79.64±0.83** | 75.70±0.67 | 73.73±0.16 | 73.11±0.25 | 79.11±1.74 | 75.44±0.50 | 78.10±0.39 | 75.38±0.27 |
| Hyperplane High Gradual Drift | **91.85±0.03** | 91.56±0.03 | 75.78±0.11 | 91.27±0.53 | 87.69±3.35 | 75.73±0.16 | 71.95±0.83 | 84.88±0.13 |
| Moving RBF | **88.00±0.34** | 86.82±0.17 | 85.11±0.70 | 67.23±0.31 | 83.20±2.35 | 51.27±0.32 | 49.62±0.67 | 39.36±0.40 |
| Moving Squares | 98.21±0.35 | **98.61±0.20** | 88.69±0.15 | 88.34±0.69 | 87.12±3.02 | 59.59±1.98 | 75.44±1.10 | 80.75±0.48 |
| Sea High Abrupt Drift | 88.52±0.03 | 88.04±0.02 | 84.33±1.31 | 88.78±0.23 | 88.60±0.23 | **88.92±0.03** | 84.39±3.45 | 88.80±0.02 |
| Sea High Mixed Drift | 87.87±0.02 | 87.47±0.03 | 84.13±1.42 | **88.27±0.38** | 87.49±0.34 | 88.03±0.11 | 81.73±3.20 | 88.13±0.03 |
| Synth Random RBF Drift | **67.48±0.14** | 65.34±0.20 | 60.78±2.86 | 65.59±0.20 | 60.46±3.54 | 62.82±1.25 | 56.22±0.77 | 55.03±0.28 |
| Synth Agrawal | 99.17±0.38 | 94.74±1.11 | 99.85±0.13 | **99.97±0.59** | 99.94±0.01 | 97.77±0.97 | 98.19±1.57 | 99.58±0.00 |
| **Avg. Accuracy** | **86.05** | 85.00 | 81.39 | 80.91 | 83.84 | 77.79 | 77.71 | 76.27 |
| **Avg. Rank** | **2.64** | 3.64 | 5.07 | 4.50 | 3.64 | 4.79 | 6.00 | 5.71 |

Table 4: Run time comparison.

| Datasets | ASML_E | ASML_B | AutoClass | OAML | EAML | ARFC | SRPC | HATC |
|---|---|---|---|---|---|---|---|---|
| Adult | 267.65 | 162.93 | 642.18 | 94.82 | 624.77 | 83.80 | 188.41 | **46.76** |
| Electricity | 118.29 | 88.09 | 621.69 | 2444.21 | 281.31 | 53.31 | 142.16 | **27.20** |
| Forest Cover | 8243.77 | 3783.63 | 12000.87 | 19989.76 | 6921.74 | **1225.47** | 6717.35 | 1751.64 |
| Insects | 977.70 | 422.64 | 1782.64 | 798.73 | 1552.31 | **113.82** | 626.20 | 124.69 |
| New Airlines | 2674.95 | 1772.96 | 5311.56 | 3145.30 | 3404.49 | 749.64 | 1824.01 | **324.86** |
| Shuttle | 693.62 | 328.57 | 504.78 | 400.50 | 499.64 | 62.87 | 150.59 | **49.25** |
| Vehicle Sensit | 4028.79 | 1548.14 | 3312.30 | **201.94** | 9193.06 | 283.39 | 2090.46 | 434.71 |
| Hyperplane High Gradual Drift | 1335.77 | 1001.76 | 4229.89 | 4081.31 | 3137.78 | 874.79 | 1913.36 | **391.91** |
| Movingrbf | 1544.65 | 876.98 | 1862.36 | 2554.01 | 2001.40 | 340.84 | 1084.49 | **186.07** |
| Moving Squares | 681.17 | 442.13 | 2252.66 | 11383.02 | 1069.37 | 264.43 | 271.95 | **93.83** |
| Sea High Abrupt Drift | 921.49 | 739.42 | 5975.26 | 964.44 | 2429.76 | 876.42 | 1404.49 | **272.97** |
| Sea High Mixed Drift | 904.30 | 725.69 | 5525.22 | 1555.90 | 2386.58 | 823.21 | 1397.29 | **278.84** |
| Synth Randomrbfdrift | 929.13 | 336.47 | 1377.71 | 190.69 | 869.08 | 124.45 | 266.35 | **66.92** |
| Synth Agrawal | 765.33 | 451.99 | 1042.93 | 256.07 | 657.41 | 96.01 | 271.52 | **50.42** |
| **Avg. Time (Sec)** | 1720.47 | 905.81 | 3317.29 | 3432.91 | 2502.05 | 426.60 | 1310.62 | **292.86** |
| **Avg. Rank** | 5.57 | 3.43 | 7.43 | 5.43 | 6.50 | 2.00 | 4.36 | **1.29** |

Table 5: Memory usage comparison.

| Datasets | ASML_E | ASML_B | AutoClass | OAML | EAML | ARFC | SRPC | HATC |
|---|---|---|---|---|---|---|---|---|
| Adult | 10.50 | 9.02 | 254.03 | 9.06 | 52.62 | 19.10 | 24.84 | **3.55** |
| Electricity | 5.12 | 5.36 | 43.00 | 18.26 | 4.82 | 6.11 | 14.90 | **2.36** |
| Forest Cover | 64.94 | 58.43 | 81.05 | 48.90 | 64.17 | 37.50 | 90.74 | **36.04** |
| Insects | 15.67 | 11.02 | 229.21 | 17.45 | 13.48 | 5.22 | 8.33 | **3.62** |
| New Airlines | 60.99 | 51.66 | 329.98 | 62.40 | 61.68 | 44.93 | 64.47 | **31.49** |
| Shuttle | 16.03 | 10.29 | 9.07 | 8.31 | 5.74 | 4.19 | 5.99 | **2.93** |
| Vehicle Sensit | 17.60 | 17.14 | 193.22 | 45.31 | 386.01 | 44.39 | 163.11 | **10.81** |
| Hyperplane High Gradual Drift | 34.45 | 33.08 | 36.66 | 56.07 | 67.12 | 98.19 | 128.39 | **29.53** |
| Movingrbf | 14.48 | 13.95 | 78.75 | **4.28** | 14.19 | 16.48 | 24.48 | 10.64 |
| Moving Squares | 16.64 | 15.91 | 14.34 | **8.79** | 12.63 | 17.28 | 10.43 | 10.24 |
| Sea High Abrupt Drift | 34.16 | 34.08 | 574.58 | 75.53 | 86.01 | 103.16 | 117.82 | **29.56** |
| Sea High Mixed Drift | 35.90 | 34.37 | 743.76 | 107.24 | 95.72 | 85.57 | 109.90 | **28.62** |
| Synth Randomrbfdrift | 10.43 | 9.63 | 26.31 | 16.35 | 14.59 | 12.59 | 16.91 | **4.94** |
| Synth Agrawal | 21.89 | 15.28 | 58.45 | 11.37 | 7.92 | 8.10 | 33.69 | **5.18** |
| **Avg. Memory (M.B.)** | 25.63 | 22.80 | 190.89 | 34.95 | 63.34 | 35.91 | 58.14 | **14.97** |
| **Avg. Rank** | 4.57 | 3.43 | 7.21 | 4.50 | 4.71 | 4.29 | 6.14 | **1.14** |

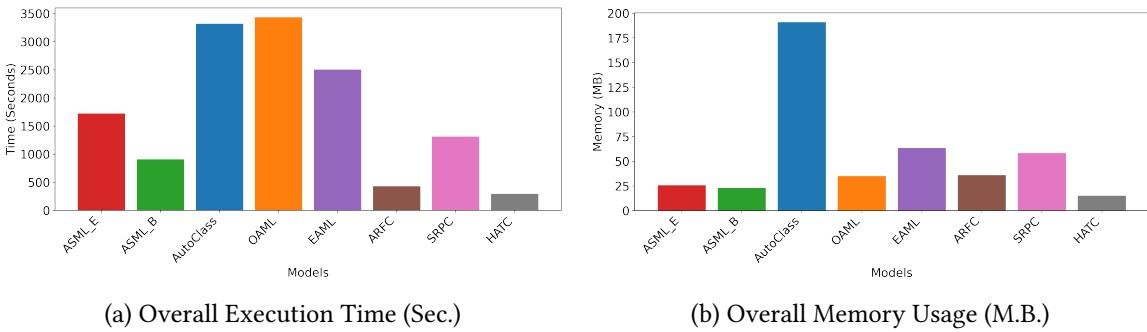

(a) Overall Execution Time (Sec.)  (b) Overall Memory Usage (M.B.)

Figure 3: Comparative chart of ASML against baseline regarding execution time and memory usage across datasets.

comparison (Figure 4b) reveals that ASML_B is ranked higher than most other algorithms and has a comparable execution time to the online learning algorithm. This indicates that ASML_B is faster than the existing AutoML systems but not significantly different from ASML_E and OAML. For memory usage (Figure 4c), ASML_B has a clear advantage over the other algorithms, as it uses remarkably less memory than ASML_E and most of the other AutoML systems.

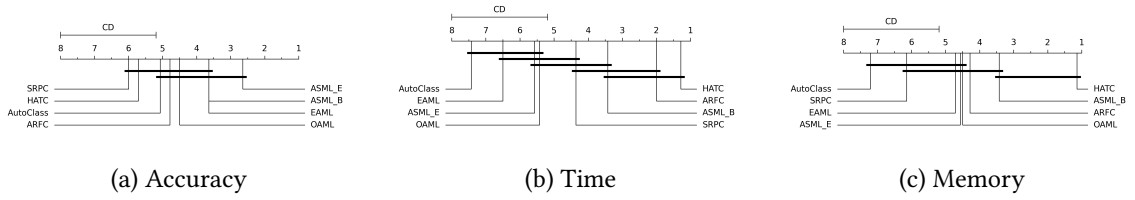

(a) Accuracy  (b) Time  (c) Memory

Figure 4: Critical differences for Accuracy, Time, and Memory

## 6.2 Discussion

This discussion highlights ASML efficiency and effectiveness over baseline AutoML systems due to several contributing factors: Firstly, ASML employs feature selection methods that reduce feature dimensions, accelerating the learning process and enhancing the model's generalization ability by eliminating irrelevant or redundant features. This capability is absent in baseline AutoML solutions. Secondly, ASML versions adopt a more efficient learning strategy. They train every pipeline exactly once with every example. In constrast, both AutoClass and EAML use online bagging, based on a Poisson distribution with a mean of 6, which in the current River implementation makes them train on every single example 6 times on average. OAML on the other hand regularly pauses stream processing to recompute the best pipeline in an offline way. This is triggered either by concept drift or after a preset number of examples. Thirdly, selecting algorithms and configurations while searching in a stream impacts the performance and efficiency of the Online AutoML systems. ASML maintains a diverse set of models by directly replacing pipelines within the ensemble, ensuring frequent model changes. This method eliminates the need for explicit diversity checks within the ensemble, as the constant introduction of new models naturally provides enough diversity to enhance performance. Lastly, ASML's search strategy differs from that of baseline AutoML systems. ASML follows an 'explore first, then select' approach, which involves exploring various models and selecting the best one to be placed in a separate space for the best model or ensemble pool. This differs from EAML, which replaces the worst model with a mutation of the current best one in the same ensemble pool, and AutoClass, which uses a similar strategy. OAML, on the other hand, initiates an offline search when a drift is detected, pausing the online learning process.

## 7 Conclusion and Future Works

In this paper, we presented Automated Streaming Machine Learning (ASML), a novel automated machine learning system for non-stationary data streams. ASML introduces two innovative features: a continuous search algorithm based on adaptive random directed nearby search (ARDNS) and a dynamic ensemble method that selects the best models for each data chunk. These features enable ASML to explore a large and diverse pipeline configuration space and leverage the data stream's feedback. We conducted comprehensive experiments on real-world and synthetic data streams with different types of drifts and compared ASML with state-of-the-art online learning algorithms. Our results show that ASML achieves better or comparable predictive performance while being more efficient regarding time and memory consumption.

In future work, we plan to extend ASML to support other machine learning tasks common in data stream applications such as anomaly detection, regression, clustering and recommendation. We believe that ASML is a versatile and powerful tool that can advance the state-of-the-art in AutoML for data streams.

## 8 Broader Impact Statement

After careful reflection, the authors have determined that this work presents no notable negative impacts to society or the environment.

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

## A  Supplemental Section

### A.1  Experiments with Artificial Data

We evaluated the performance of ASML on different types of concept drift scenarios using synthetic datasets. We found that ASML was very effective in adapting to gradual drifts, as it achieved the highest accuracy on Hyperplane High Gradual Drift (Figure 6a) and Moving RBF (Figure 6b) datasets. These datasets simulate smooth and continuous changes in the data distribution over time. ASML also handled predictable gradual changes well, as it outperformed other methods on the Moving Squares dataset (Figure 6c). This dataset mimics a scenario where the class boundaries shift periodically in a cyclic manner. Moreover, ASML was competitive on abrupt drift datasets, such as Sea High Abrupt Drift (Figure 6d), where the data distribution changes suddenly and randomly. However, it did not surpass some specialized methods that are designed for this type of drift. Interestingly, ASML showed the best performance on the highly dynamic Random RBF Drift dataset (Figure 6f), where the data distribution changes frequently and unpredictably. This demonstrates the robustness of ASML in volatile environments. On the other hand, ASML was not the clear winner on complex mixed drift datasets, such as Sea High Mixed Drift (Figure 6e), where the data distribution changes both gradually and abruptly. It was also not the best performer on synthetic data, such as Agrawal (Figure 6g), where the data distribution is generated by a predefined function. However, ASML was still competitive and close to the top performers on these datasets. Overall, ASML exhibits state-of-the-art adaptability across different modes of concept drift.

### A.2  Experiments with Real-World Data

We tested ASML on real-world datasets that reflect various types of concept drift in practical settings. We found that ASML performed well on the Adult dataset (Figure 7a), where the data distribution changes due to socio-economic factors. It matched the accuracy of the best performing methods on this dataset. We also found that ASML excelled on the Electricity (Figure 7b) and Forest Cover (Figure 7c) datasets, where the data distribution changes due to environmental factors. It surpassed all baseline algorithms on these datasets, showing its real-world effectiveness. However, ASML faced more challenges on the Insects (Figure 7d) and Vehicle SensIT (Figure 7g) datasets, where the data distribution changes due to seasonal factors. It achieved strong accuracy on these datasets. ASML was not the clear winner on the New Airlines (Figure 7e) and Shuttle (Figure 7f) datasets, where the data distribution changes due to operational factors. It was still competitive and close to the top performers on these datasets. Overall, ASML demonstrated state-of-the-art adaptability across different real-world scenarios of concept drift.

### A.3  Pipeline Analysis

This section analyzes how ASML dynamically changes its machine learning pipeline to cope with different datasets. ASML can select the most suitable machine learning algorithm based on the changing characteristics of the incoming data stream. We visualized the algorithm selection for four datasets: *Electricity* (Figure 8a), *Insects* (Figure 8b), *Moving Square* (Figure 8c), and *Synthetic Random RBF Drift* (Figure 8d). These visualizations demonstrate how ASML adapts its algorithm selection strategy dynamically in response to different data patterns in streaming settings.

A prime example of ASML's adaptability is observed in the Shuttle dataset (Figure 8e), where ASML's algorithm performance tracking reveals a sophisticated selection process. Initially, the system starts with a HoeffdingTreeClassifier paired with a StandardScaler, achieving an accuracy of 97.34%. As the data stream progresses, ASML identifies that a pipeline with MinMaxScaler, SelectKBest, and AdaptiveRandomForestClassifier is more effective, reaching an accuracy of 98.65%. Towards the end of the data stream, the system further optimizes its choice, settling on a combination of StandardScaler and AdaptiveRandomForestClassifier, which delivers the highest accuracy of 99.04%. This example illustrates ASML's capability to monitor and adapt its pipeline choices over time, ensuring the selection of the most effective algorithmic strategies for the Shuttle dataset.

### A.4 Ablation Study

We conducted an ablation study to investigate the impact of different settings of ASML on its performance. We focused on two datasets: *Adult* (Figure 12e) and *Vehicle SensIT* (Figure 12f). We found that tuning the hyperparameters and selecting the features were crucial for enhancing ASML's performance. Hyperparameter tuning improved the model's accuracy significantly over time, as it adjusted the model's parameters to fit the data better. Feature selection, which selected only the most relevant data for training, also boosted ASML's accuracy, as it reduced the noise and redundancy in the data. These visualizations show that both hyperparameter tuning and feature selection helped ASML adapt better to the data and predict more accurately. However, we note that these results may not generalize to all datasets.

The ablation study further investigated the impact of different hyperparameter settings on ASML's performance across two datasets, Electricity and RandomRBFDrift, for their distinct characteristics representing real-world and synthetic data, respectively. enabling a rigorous evaluation of ASML's adaptability under varying conditions. The findings were visually represented in a series of figures: Figure 10 for accuracy scores, Figure 11 for computational time, and Figure 12 for memory usage. In selecting the optimal values for ensemble size, exploration window, and budget size in ASML, we navigate a delicate trade-off among model performance, computational time, and memory usage. The ensemble size directly impacts the model's predictive power and stability, with larger ensembles typically providing more diversity. However, increasing ensemble size also escalates computational demands, leading to longer training times and higher memory consumption. Thus, we strive to strike a balance where the ensemble size is sufficiently large to capture diverse perspectives while avoiding excessive computational overhead. Similarly, the exploration window size influences ASML's ability to adapt to concept drifts and evolving data distributions. Thus, we aim to choose an exploration window size that balances the need for accurate adaptation to data dynamics with computational efficiency. Moreover, the budget size dictates the resources allocated to hyperparameter optimization and model adaptation. A larger budget allows for more extensive exploration of hyperparameter space. However, allocating a larger budget comes at the cost of increased computational time and memory usage, as more iterations of optimization algorithms are required. Hence, we carefully consider the budget size to ensure efficient utilization of computational resources while maximizing model performance. However, these tradeoffs may varies for different datasets and not be generalized to all datasets.

### A.5 Predictive Performance Comparison Under Same Configuration Settings

We compared the predictive performance of ASML_E with the baseline online AutoML algorithms under the same configuration space (Table 6). This allows us to assess the methods fairly and objectively. We also tried to make algorithm parameters exactly as similar as possible. We found that ASML outperformed the other methods on average test accuracy, time, and memory. ASML achieved an average test accuracy of 85.44%, which was higher than EvoAutoML's 83.82%, OnlineAutoML's 75.09%, and AutoClass's 83.87% (Table 7). ASML also ran faster and used less memory than the other methods. It took an average time of 1114.38 seconds and an average memory of 15.71 MB, while EvoAutoML took 2142.58 seconds and 63.13 MB, OnlineAutoML took 1387.67 seconds and 53.30 MB, and AutoClass took 1125.14 seconds and 20.79 MB (Table 8 and Table 9). Moreover, ASML had the best average rank of 1.57 across all metrics, followed by EvoAutoML with 2.21, AutoClass with 2.86, and OnlineAutoML with 3.36. The critical difference diagram (Figure 5) also showed that ASML was remarkably better than the other methods. Therefore, we conclude that ASML has superior predictive performance over the baseline online AutoML algorithms under the same configuration.

Table 6: Configuration space used for comparing AutoML system in the same configuration

| Algorithm | Hyperparameter | Search Range |
|---|---|---|
| LogisticRegression | l2 | [0.0, 0.01, 0.001] |
| HoeffdingTreeClassifier | max_depth | [10, 30, 60] |
| | grace_period | [10, 100, 200] |
| | max_size | [5, 10] |
| KNNClassifier | n_neighbors | [1, 5, 20] |
| | window_size | [100, 500, 1000] |
| | weighted | [True, False] |
| GaussianNB | | |
| MinMaxScaler | | |
| StandardScaler | | |
| MaxAbsScaler | | |

Table 7: Accuracy of Different Models on Datasets

| Datasets | ASML_E | EAML | OAML | AutoClass |
|---|---|---|---|---|
| Adult | **81.24** | 79.98 | 66.97 | 81.04 |
| Electricity | **89.06** | 88.95 | 81.32 | 85.39 |
| Forest Cover | 95.54 | 94.17 | **96.53** | 95.30 |
| Insects | **71.65** | 69.50 | 66.07 | 65.23 |
| New Airlines | 66.26 | **67.60** | 65.71 | 65.15 |
| Shuttle | **99.08** | 98.59 | 84.37 | 96.04 |
| Vehicle Sensit | 78.78 | **79.59** | 74.63 | 75.38 |
| Hyperplane High Gradual Drift | **91.83** | 86.36 | 91.74 | 89.03 |
| Moving RBF | **88.14** | 83.66 | 50.77 | 83.12 |
| Moving Squares | 93.31 | 89.60 | 72.24 | **99.66** |
| Sea High Abrupt Drift | 88.40 | **88.50** | 88.49 | 88.18 |
| Sea High Mixed Drift | **87.84** | 87.49 | 79.15 | 87.24 |
| Synth Random RBF Drift | **65.28** | 59.54 | 57.48 | 63.50 |
| Synth Agrawal | 99.77 | **99.94** | 75.84 | 99.92 |
| **Avg.** | **85.44** | 83.82 | 75.09 | 83.87 |
| **Avg. Rank** | **1.57** | 2.21 | 3.36 | 2.86 |

Table 8: Time (Sec.) Taken by Different Models on Datasets

| Datasets | ASML_E | EAML | OAML | AutoClass |
|---|---|---|---|---|
| Adult | 217.56 | 511.01 | **182.60** | 233.17 |
| Electricity | 156.19 | 262.83 | **77.47** | 148.88 |
| Forest Cover | 6649.63 | 5906.38 | 9052.29 | **2722.66** |
| Insects | 601.38 | 1005.55 | **368.27** | 1846.63 |
| New Airlines | **1419.91** | 2451.33 | 2137.76 | 3484.99 |
| Shuttle | 288.88 | 428.69 | 3358.21 | **231.69** |
| Vehicle Sensit | 1895.98 | 8410.68 | **718.00** | 2381.86 |
| Hyperplane High Gradual Drift | 912.57 | 2833.93 | **410.39** | 1143.96 |
| Moving RBF | 919.75 | 1626.91 | **139.56** | 825.63 |
| Moving Squares | 393.70 | 1206.61 | **326.08** | 427.89 |
| Sea High Abrupt Drift | 742.52 | 1999.30 | **505.61** | 686.56 |
| Sea High Mixed Drift | **663.76** | 2107.04 | 2022.44 | 695.08 |
| Synth Random RBF Drift | 469.04 | 663.34 | **62.43** | 426.53 |
| Synth Agrawal | 270.43 | 582.52 | **66.30** | 496.42 |
| **Avg.** | **1114.38** | 2142.58 | 1387.67 | 1125.14 |
| **Avg. Rank** | 2.21 | 3.64 | **1.64** | 2.50 |

Table 9: Memory Usage (M. B.) by Different Models on Datasets

| Datasets | ASML_E | EAML | OAML | AutoClass |
|---|---|---|---|---|
| Adult | **4.16** | 53.98 | 14.34 | 8.42 |
| Electricity | 3.03 | **0.74** | 3.20 | 7.31 |
| Forest Cover | 44.48 | **11.23** | 273.18 | 43.66 |
| Insects | **5.13** | 6.93 | 35.52 | 22.99 |
| New Airlines | 33.45 | **8.52** | 18.52 | 63.80 |
| Shuttle | 3.80 | **0.21** | 4.47 | 5.00 |
| Vehicle Sensit | **8.28** | 420.76 | 17.94 | 21.46 |
| Hyperplane High Gradual Drift | 29.00 | **19.50** | 114.46 | 33.68 |
| Moving RBF | 11.60 | **8.69** | 40.00 | 16.26 |
| Moving Squares | **11.03** | 183.04 | 43.37 | 13.82 |
| Sea High Abrupt Drift | 26.26 | 87.82 | 113.97 | **22.57** |
| Sea High Mixed Drift | 26.28 | 74.56 | 64.45 | **11.91** |
| Synth Random RBF Drift | 6.63 | 7.09 | **0.60** | 11.52 |
| Synth Agrawal | 6.87 | **0.66** | 2.15 | 8.66 |
| **Avg.** | **15.71** | 63.13 | 53.30 | 20.79 |
| **Avg. Rank** | **1.93** | 2.21 | 3.00 | 2.86 |

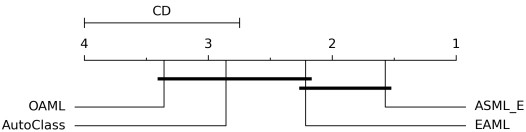

Figure 5: Critical Difference Diagram of Accuracy with same Configuration

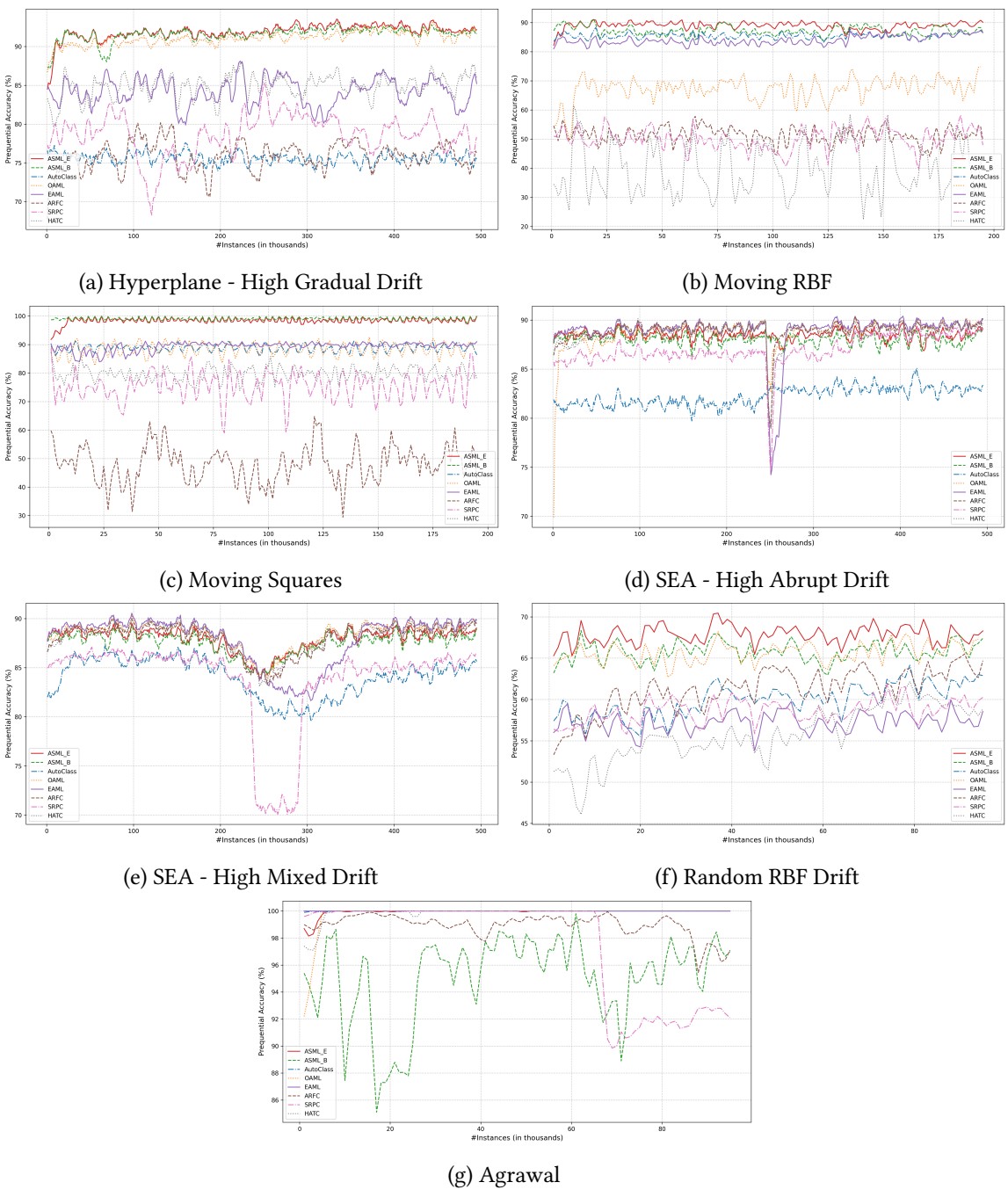

(a) Hyperplane - High Gradual Drift

(b) Moving RBF

(c) Moving Squares

(d) SEA - High Abrupt Drift

(e) SEA - High Mixed Drift

(f) Random RBF Drift

(g) Agrawal

Figure 6: ASML performance on various synthetic datasets

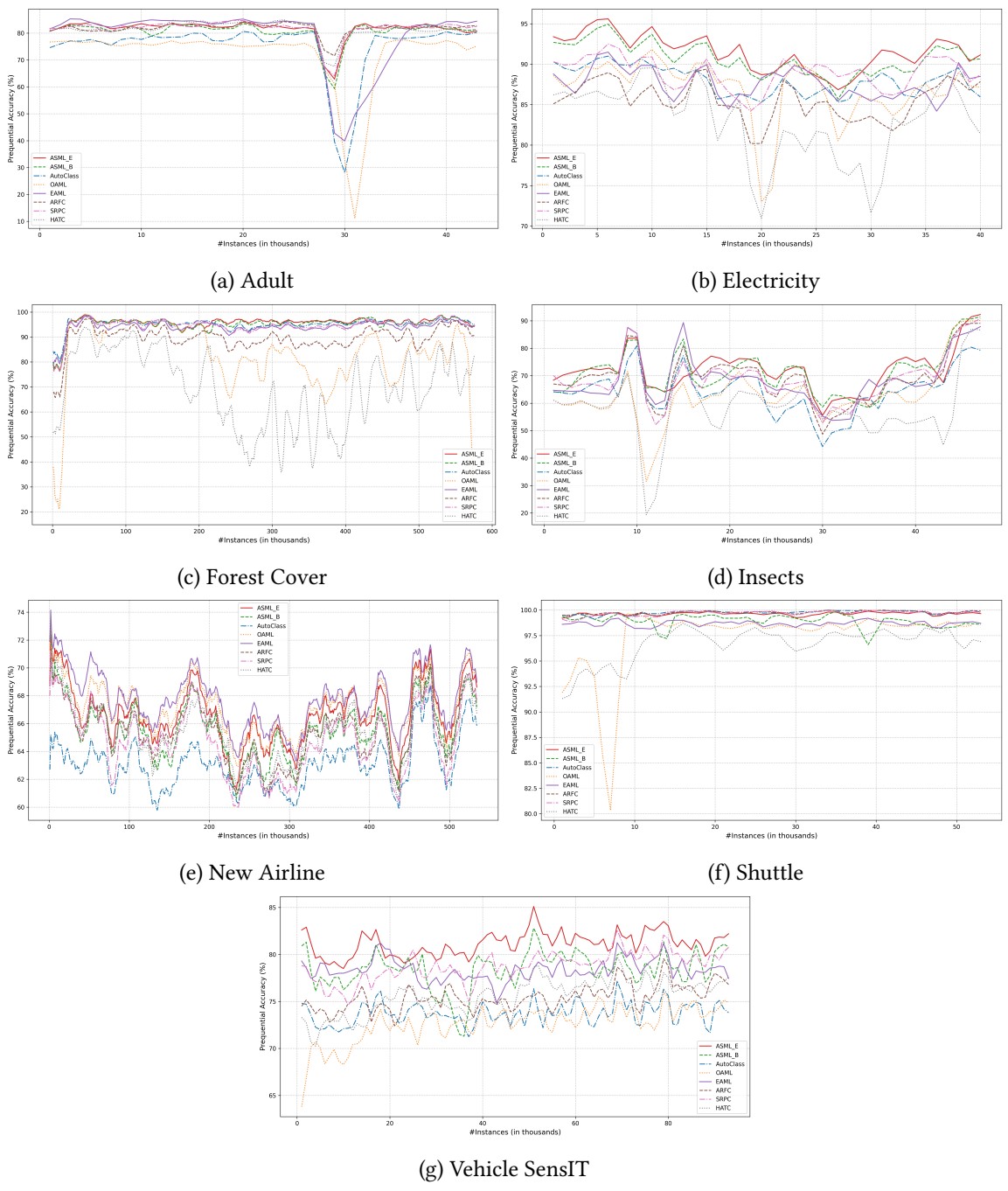

(a) Adult

(b) Electricity

(c) Forest Cover

(d) Insects

(e) New Airline

(f) Shuttle

(g) Vehicle SensIT

Figure 7: ASML performance on various real-world datasets

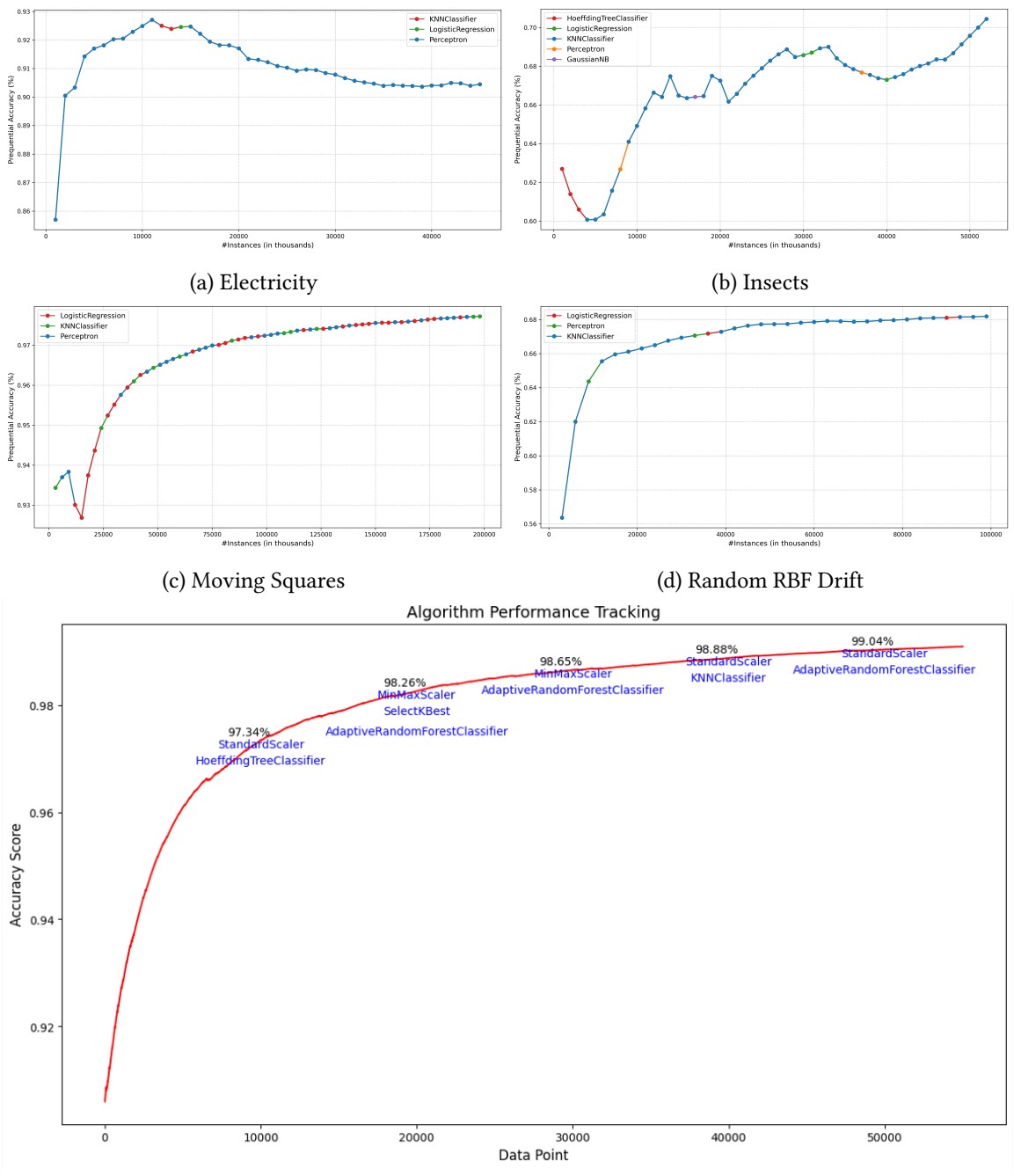

(a) Electricity

(b) Insects

(c) Moving Squares

(d) Random RBF Drift

(e) Pipeline selection tracking in ASML on the Shuttle

Figure 8: Algorithm selection tracking in ASML on various datasets

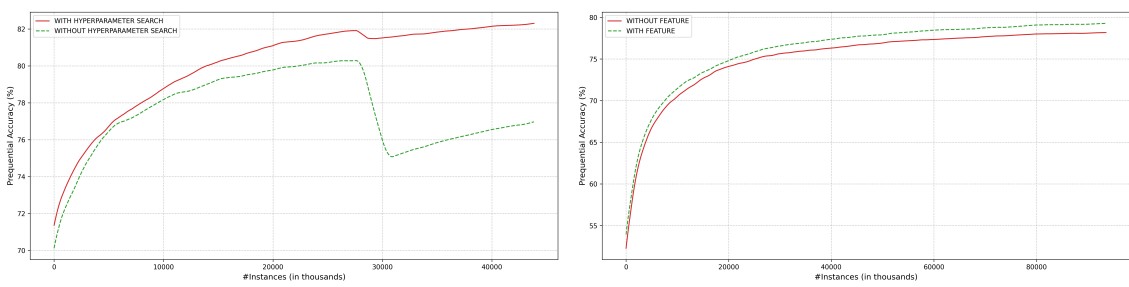

(a) With and without hyperparameter tuning       (b) With and without feature selection

Figure 9: ASML performance enhancements on different aspects

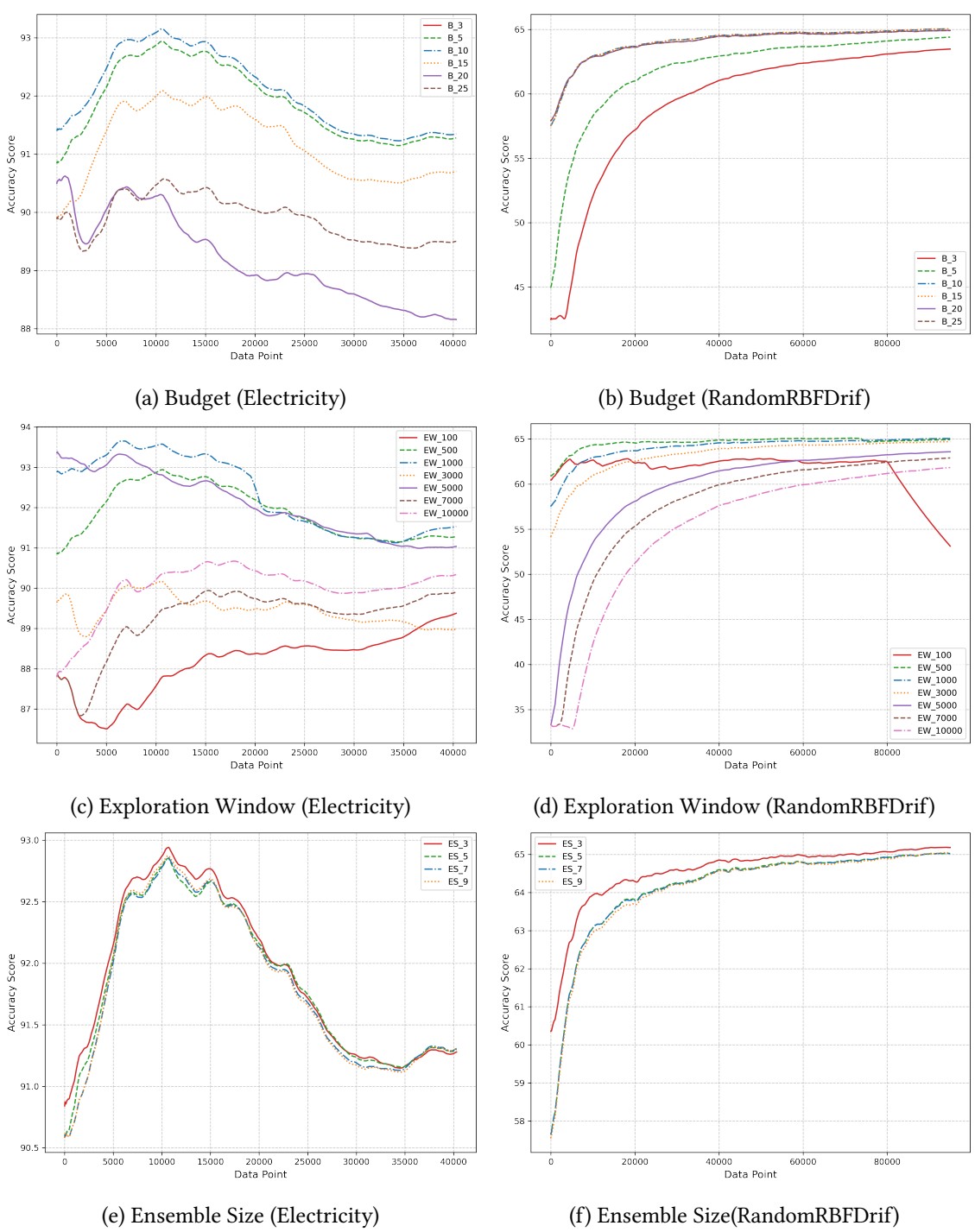

Figure 10: Performance (Measured in Prequential Accuracy): An Ablation Study on the Impact of Different Hyperparameters of ASML

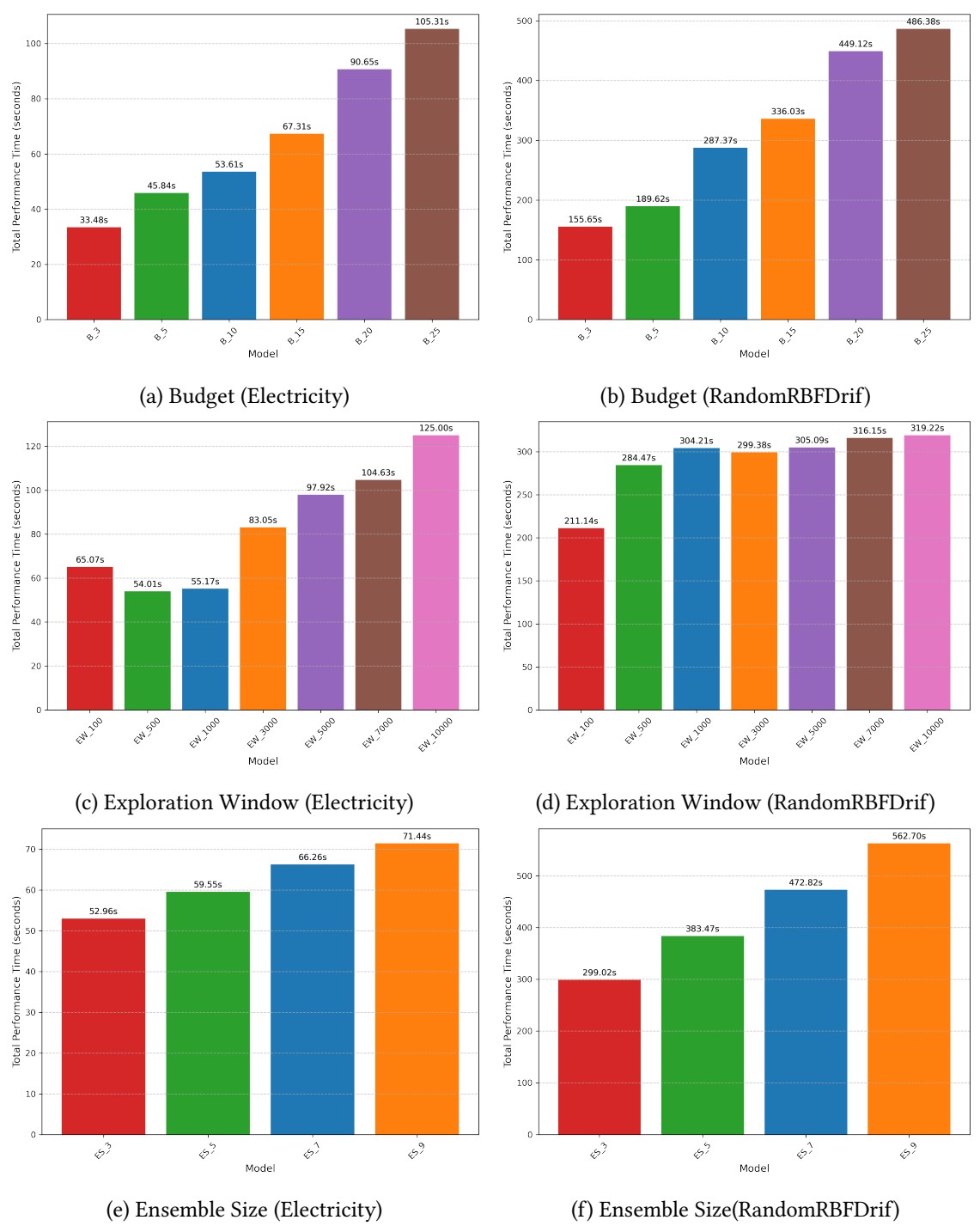

Figure 11: Time Taken (in Seconds): An Ablation Study on Various Hyperparameters of ASML

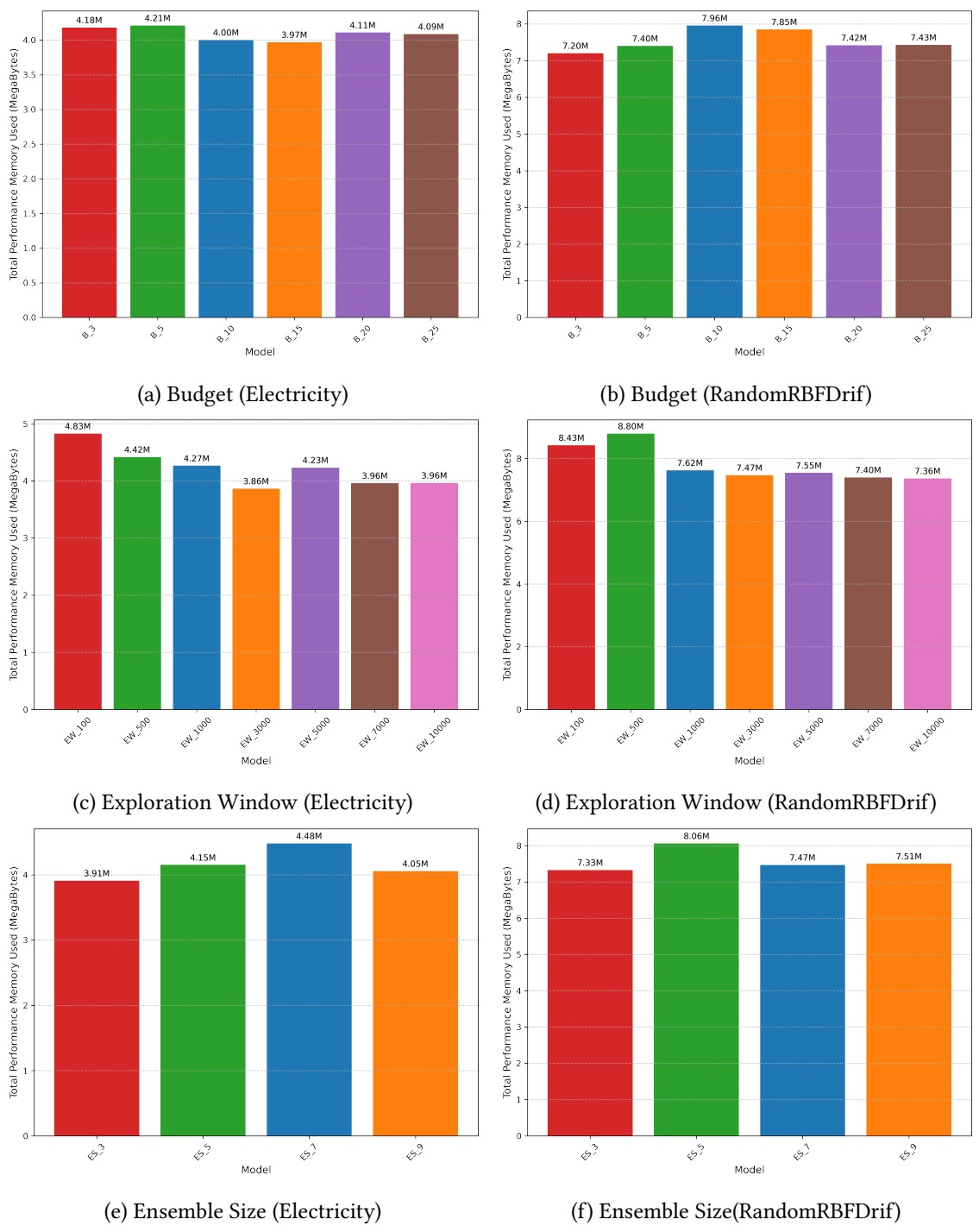

Figure 12: Memory Usage (in Megabytes): An Ablation Study on Various Hyperparameters of ASML

