# OpenReview forum: "ASML: A Scalable and Efficient AutoML Solution for Data Streams"
_automl.cc/AutoML/2024/Conference — AutoML 2024_

### Official Review · Reviewer_h64R · 2024-03-24

**Potential Impact On The Field Of Automl Rating:** 3
**Technical Quality And Correctness Rating:** 3
**Clarity Rating:** 3

**Summary Of Contributions:**

This paper proposes a novel framework for performing full pipeline optimization of machine learning methods; specifically, their algorithm optimizes the preprocessing, feature engineering, model and all associated hyperparameters. The proposed framework uses a combination of random search and ARDNS to maintain an ensemble of high-performing pipelines. The authors empirically evaluate their method and find that it outperforms (but not statistically significantly) other alternative automated machine learning methods. They also evaluated the computational cost of their method relative to these alternatives and again found that it was competitive but not notably better than the baselines.

**Actions Required To Increase Overall Recommendation:**

I would like to see an extended ablation analysis on the entire set of datasets considered in the main text that also explores the impact on computational costs, not just the prequential accuracy.

Some discussion about the limitations/potential extensions I discussed in the "Technical Quality And Correctness" section would also be welcome.

**Clarity:**

For the most part, the document is clear and well-written. If I had to pick nits, I would say that the figures are all a bit small given the size of the text within (only relevant for those of us who enjoy reviewing on paper, I suppose). Also, the notation in equation (2), $\{\mathbf{X}_t, \mathbf{y}_t\}$ is not defined before its use.

Also, this is potentially a misunderstanding of how ARDNS works on my part but shouldn't the value of H3 in the "New Config" shown in Figure 2 be 0.3, not 0.2? Based off of Algorithm 2, it appears that each hyperparameter should move in the same direction and in Figure 2, both H1 and H2 went to the "Lower" branch whereas H3 seems to have gone down the "Random" branch.

**Overall Review:**

The primary strengths of this paper in my opinion are the simplicity/flexibility of the main idea and the rigor of the empirical evaluations. In particular, i want to highlight the ablation study performed by the authors in the appendix as an interesting and important aspect of this work (one that I think should be expanded and ideally included in the main text).

In terms of weaknesses, the proposed method does have some rather major limitations (see my comments in the "Technical Quality And Correctness" section) and the empirical results are on the weaker side: neither of the proposed methods consistently or statistically significantly outperforms the baselines on any of the presented metrics.

**Potential Impact On The Field Of Automl:**

Certainly the proposed work addresses a relevant problem to the field and in fact, approaches it from a full pipeline perspective where many methods simply consider the model selection and/or hyperparameter optimization aspects of the problem. Additionally, the proposed method is intuitive, seeming simple to implement, and flexible to various preprocessors, feature selectors, and classifiers, all of which serve to broaden broaden the potential impact of this work.

**Review Confidence:**

3

**Review Rating:**

7

**Review Summary:**

The proposed idea is simple, intuitive and addresses a problem relevant to the AutoML community, which outweighs the primary concern of somewhat weak empirical findings.

**Technical Quality And Correctness:**

The proposed methods, experimental setup and statistical analysis appear to be technically sound. In terms of their rigor/quality, I do have some concerns. In Algorithm 1, the fact that initialization requires the algorithm to process every combination of preprocessors, feature selectors and classifiers seems excessive and might present a bottleneck for scaling the proposed method to broader sets of those pipeline components; it seems like it might have been sufficient to seed the algorithm with a randomly selected set of pipelines instead of having to consider all possible pipelines.

Another (somewhat related) limitation appears to be that the proposed method can only handle continuous hyperparameters by discretizing the search space (an admittedly common strategy/approximation in the AutoML space). It seems that the proposed method could have been relatively easily expanded/adapted to handle continuous parameters as random sampling could be used to generate real-values for the continuous hyperparameters (perhaps bounded over some domain) and ARDNS can be handle continuous search domains simply by specifying a step size (or even randomly sampling a step size) for the upper and lower directions.

---

### Official Review · Reviewer_WmdB · 2024-03-25

**Potential Impact On The Field Of Automl Rating:** 3
**Technical Quality And Correctness Rating:** 3
**Clarity Rating:** 3
**Actions Required To Increase Overall Recommendation:** 1. Address the performance metric in …

**Summary Of Contributions:**

The paper proposes a novel approach towards AutoML for online learning. ASML is designed for data streams, that canm often suffer from data drift and changing characteristics. The framework is stated to be adaptive, and the authors use a strategy that explores pipeline configuration.
The authors provide a good exposition of current literature with current works such as EvoAutoML, AutoClass and OAML. They also proveide a a comprehensive coverage on other standard AutomML frameworks.
The authors formulate this as a dynamic CASH problem., and their search space consists of pipeline and hyperparmeters.

**Clarity:**

The writing style is overall fluid, and authors doa  good job of explaining their design choices.
1. Figure 1 is not legible.
2. Figure 2 caption is not explanatory. Again the text size is rather small.

**Overall Review:**

Positive:
- ASML is a novel framework
- Literature review is comprehensive
- The paper is well written, and  clarity is good.
- Experimental section is well presented. Both aspects --- time and memory are considered.

Negative:
-  A concern here is that the framework is dependent on River, which is functional but does not seem nature and well maintained.
- The pipeline search at each W timesteps, with a window of size W can be expensive --- if W is small. Why not employ a data drift detection mechanism? Approaches  such as Bayesian Change Point Detection are comparatively cheaper.
- The experimental results are not very convincing. Table 3 --- ASML_E is the best performer overall, but ASML_B ties with EAML in terms of rank.  Also ASML is not the most memory efficient or quickest (and that is fine). But ASML_E does not seem to provide that strong of an advantage over ASML_B--- which is a bit counterintuitive.

**Potential Impact On The Field Of Automl:**

This is a useful paper to the AutoML research community. Online learning and continual learning are challenging, especially data streams.

**Review Confidence:**

3

**Review Rating:**

7

**Review Summary:**

The paper is novel and interesting. It presents a framework, that solves a challenging problem of some practical importance. The paper is a well written work and is technically sound.

**Technical Quality And Correctness:**

The pipeline search is partially random. While exploration and exploration are important, it is not clear if this randomness could be replaced with a more directed approach. Also seems at a budget of B -- there are B+1 condidates?  Also it would be interesting to explore if the search space could be pruned.

The idea of hyperparameters space can be searched based on a directional(grid) search with a mixture of randomness in ARDNS algorithm (Algo 2) looks intuitively correct. It may however be slow on cases where hyperparam count is high. Joint hyperparameter optimization is a challenging task.

Tha experimental section is well presented, with a detailed analysis. The only concern is the performance imetric used.

---

### Official Review · Reviewer_c8yA · 2024-04-01

**Potential Impact On The Field Of Automl Rating:** 4
**Technical Quality And Correctness Rating:** 2
**Clarity Rating:** 3

**Summary Of Contributions:**

The contributions of this paper are:
1. A new framework for online AutoML on data streams.
2. An adaptive search strategy based on random-directed nearby search.
3. Experiments showing this method beats SOTA in terms of performance and resource efficiency.

**Actions Required To Increase Overall Recommendation:**

The main actions I would need to improve my rating would be:
1. Clarify how hyperparameters are set for the proposed method.
2. Report results on ablations on hyperparameters such as ensemble size, window size and budget.
3. Ensure that the experimental comparisons are fair, e.g. that all methods are allowed to optimise their own hyperparameters on these datasets.
4. Improve the clarity of the text in several places as mentioned.

**Clarity:**

The Related Works section discusses some previous online AutoML methods. However, the descriptions of OAML, EvoAutoML and AutoClass sound extremely similar (all using some evolutionary approach with mutations). It would be better to mention them as population-based methods and then more clearly outline their differences.

The end of the same section is too vague. Which methods have which limitations? How are these limitations causing the methods to fail? How is your method tackling them?

Fig. 1 is too small, making it hard to read the text in the figure without digital zoom.

Sometimes variables are written in lowercase and sometimes in uppercase (e.g. S and Wt).

If space is short, there is a lot of repetition that mentions the pre-processors, feature selectors, classifiers, hyperparameters etc. Some of this can be cut.

Algorithm 2 takes as input the direction to change parameter d but it is also selected on line 1.

**Overall Review:**

This paper introduces an online AutoML method that outperforms all competitors. However, it is not clear how the hyperparameters of their method are chosen and it might be an unfair comparison to competitors if their hyperparameters are fixed. This problem, along with other issues of clarity lead me to give a lower rating.

Below are some further issues:
1. The algorithm requires all configurations to be evaluated for the first window of data. This will limit the method to settings where the number of possible configurations is relatively small. Would an alternative be to initially only evaluate a smaller subset of a larger search space?
2. ARDNS in its current form requires an order in the search space. For the classifier options, this may not be natural. What do the "lower" and "upper" operations mean for a search space that includes purely categorical choices, e.g. perceptron, trees, k-NN etc.? A discussion of this should be included.

**Potential Impact On The Field Of Automl:**

Streaming data with distribution shifts are very common for real tasks and existing machine learning solutions struggle in these scenarios. This makes the topic of automated online learning over such streams very important.

**Review Confidence:**

4

**Review Rating:**

4

**Review Summary:**

This paper introduces an online AutoML method that outperforms all competitors. However, it is not clear how the hyperparameters of their method are chosen and it might be an unfair comparison to competitors if their hyperparameters are fixed. This problem, along with other issues of clarity lead me to give a lower rating.

**Technical Quality And Correctness:**

Online learning is a setting where many hyperparameters are involved and are continuously changing. While the proposed method and other competitor methods adjust the hyperparameters of the inner models, the hyperparameters of the methods themselves are not optimised for these datasets. The authors do not mention how they select their method's hyperparameters, such as ensemble size, window size and budget. They do however explicitly mention that the baselines are using default values from the original papers. In benchmarks like this, it is very important that all baselines should have their hyperparameters optimised, otherwise the results may not be fair and representative. This may be a reason why the proposed method outperforms all competitors. Numerous works have shown how poor practice leads to overclaiming and false sense of progress. See [1] and [2] for examples.

Some unbacked claims are made in the paper. E.g. "The current ensemble size of three strikes the best balance between diversity and computational demand". There seems to be no data to back this up. Ablation studies on algorithm hyperparameters such as ensemble size, window size and budget would improve this paper.

The use of statistical tests and CD diagrams is good and the focus on both performance and efficiency gives a broad picture of the methods quality.

[1] A Metric Learning Reality Check, Kevin Musgrave, Serge Belongie, Ser-Nam Lim, In ECCV, 2020.

[2] In Search of Lost Domain Generalization, Ishaan Gulrajani, David Lopez-Paz, ICLR, 2020.

---

### Meta-Review · Area_Chair_2Vs7 · 2024-04-22

**Paper Recommendation:** Accept
**Confidence:** 5

**Metareview:**

This paper proposed a simple framework for AutoML in the streaming data setting. It considers a complete search space of data preprocessing, feature selection, algorithm selection and hyperparameter optimizations. When a new batch of data arrives, the proposed method will take the best configuration from the last iteration, and split the remaining budget equally for exploration with random configuration and exploitation with Adaptive Random Directed Nearby Search (ARDNS) method. The authors compared a few relevant baselines and showed the method has competitive performance. The abalations and same configuration space experiments added in the appendix increased the soundness of the work.

The strengths of the work are the simplicity, clarity and reproducibility. The empirical results seem not particularly strong and there are areas where this work can be improved such as discretization of the search space, scalability when taking all combinations of the search space. I agree with most reviewers, the quality of writing, simplicity of the proposed algorithm and good reproducibility outweigh these weaknesses. I am inclined to accept the paper.

---

### Decision · Program_Chairs · 2024-04-29

**Decision:**

Accept

**Comment:**

Thank you for submitting your paper. We are happy to tell you that we accept your paper to the main track. See you in Paris.